# Diagnostic activity impacts lifetime risk of prostate cancer diagnosis more strongly than life expectancy

**Andri Wilberg Orrason**[1]*, **Marcus Westerberg**[2], **Peter Albertsen**[3], **Johan Styrke**[4], **David Robinson**[5], **Hans Garmo**[1], **Pär Stattin**[1]

**1** Department of Surgical Sciences, Uppsala University Hospital, Uppsala, Sweden, **2** Department of Mathematics, Uppsala University, Uppsala, Sweden, **3** Department of Surgery (Urology), UConn Health, Farmington, CT, United States of America, **4** Department of Surgical and Perioperative Sciences, Urology and Andrology, Umeå University, Umeå, Sweden, **5** Department of Urology, Region of Jönköping, Sweden

* andri.wilberg@surgsci.uu.se

**Data Availability Statement:** Data used in the present study was extracted from the Prostate Cancer Database Sweden (PCBaSe), which is based on the National Prostate Cancer Register

## Abstract

The main aim of the study was to determine the impact of diagnostic activity and life expectancy on the lifetime risk of a prostate cancer diagnosis. We used a state transition simulation model based on Swedish population-based data to simulate life trajectories for 2,000,000 men from age 40 to 100 in order to estimate the lifetime risk of a prostate cancer diagnosis. Risk estimates were determined by the level of diagnostic activity and estimated life expectancy. Higher exposure to diagnostic activity resulted in more prostate cancer diagnoses. This was especially true for men diagnosed with low or intermediate grade disease. Men exposed to high diagnostic compared to low diagnostic activity had a five-fold increased lifetime risk (22% vs. 5%) of being diagnosed with a low or intermediate-risk prostate cancer and half the risk of being diagnosed with a high-risk prostate cancer (6% vs. 13%). Men with a long life expectancy had a higher lifetime risk of a prostate cancer diagnosis both overall (21% vs. 15%) and in all risk categories when compared to men with a short life expectancy. The lifetime risk of a prostate cancer diagnosis is strongly influenced by diagnostic activity and to a lesser degree by life expectancy.

## Introduction

One in every 6 to 8 men in Western countries such as Sweden, the United Kingdom and the United States, is diagnosed with prostate cancer [1–3]. This lifetime risk estimate, however, depends upon several factors including the intensity of prostate specific antigen (PSA) testing of asymptomatic men, the availability of urologic health care including access to magnetic resonance tomography (MRT), ultrasound examinations and prostate biopsies and the number of cores obtained at biopsy. Life expectancy including chronic exposure to environmental risk factors also impacts the incidence of a prostate cancer diagnosis.

The incidence of prostate cancer increased dramatically in Sweden in the late 1990's following the introduction of PSA testing. By the early 2000's, despite national recommendations

(NPCR) of Sweden and linkage to several national health-data registers. The data cannot be shared publicly because the individual-level data contain potentially identifying and sensitive patient information and cannot be published due to legislation and ethical approval (https:// etikprovningsmyndigheten.se). Use of the data from national health-data registers is further restricted by the Swedish Board of Health and Welfare (https://www.socialstyrelsen.se/en/) and Statistics Sweden (https://www.scb.se/en/) which are Government Agencies providing access to the linked healthcare registers. The data will be shared on reasonable request in an application made to any of the steering groups of NPCR and PCBaSe. For detailed information, please see www.npcr.se/ in-english, where registration forms, manuals, and annual reports from NPCR are available alongside a full list of publications from PCBaSe. The statistical program code used for the present study analyses can be provided on request (contact npcr@npcr. se).

**Funding:** This work was supported by The Swedish Cancer Society (190030), and Uppsala County Council. These grants were unconditional, and the funding organization had no influence on the work performed.

**Competing interests:** The authors have declared that no competing interests exist.

against PSA testing, almost half of all men age 50–70 years had undergone PSA testing [4–6]. During the same period, life expectancy for Swedish men increased from 76 to 80 years [7]. As a consequence, men born in the 1930's have a lower lifetime risk of being diagnosed with prostate cancer compared to men born in the 1950's.

We previously created the Proxy-based Risk-stratified Incidence Simulation Model–Prostate Cancer (PRISM-PC), to simulate real-life scenarios of prostate cancer [8]. We have used this model to describe the interplay between age, incidence, mortality, and diagnostic activity without explicitly modelling the natural history of the disease.

The aim of this study was to quantify the effect of diagnostic activity and life expectancy on the lifetime risk of a prostate cancer diagnosis overall and by risk categories.

## Materials and methods

### Simulation model

We combined Swedish demographic data with data available in the Prostate Cancer data Base Sweden (PCBaSe) on all men diagnosed with prostate cancer from 1992 to 2016 and their prostate cancer-free comparator men, matched on birth year and region to construct a simulation model, PRISM-PC [8–10]. In brief, PCBaSe contains information on men diagnosed with prostate cancer and registered in the National Prostate Cancer Register (NPCR) that has been linked to a number of other national health care registers and demographic databases held by The National Board of Health and Welfare and Statistics Sweden [9]. We applied PRISM-PC to estimate the lifetime risk of a prostate cancer diagnosis under three different levels of diagnostic activity and life expectancy. Our PRISM-PC model has been described in detail previously [8, 11]. For a hypothetical birth cohort, the model simulates life trajectories from age 40 to age 100 for each man in the cohort. At each year the model simulates who has been diagnosed with prostate cancer, remained prostate cancer-free, died from prostate cancer or died from other causes than prostate cancer, resulting in complete life histories. For clarity, our model included life spans with an upper limit of 100 years but not all men in the simulation lived to age 100. The model is based on a proxy for the unobserved diagnostic activity defined as the age and calendar year-specific incidence of low and intermediate-risk prostate cancer in each of the 21 regions in Sweden. An age-period-cohort model was used to model the age-dependent life expectancy and trends, based on a sample of Swedish men without a prior prostate cancer diagnosis alive from 1992 to 2016 [8, 12, 13]. Model extrapolation was used to derive lifetime trends in life expectancy outside the time frame 1992–2016, so that life expectancy between ages 40–100 could be projected for men born from 1912 to 1992. Age limits were chosen since the risk of a prostate cancer diagnosis is very rare outside these age limits. In the NPCR, 25 men less than 40 years and six men older than 100 years were registered with prostate cancer from 1992 to 2016.

To stratify the risk of disease progression at diagnosis, we modified the National Comprehensive Cancer Network Categorization previously used in PCBaSe [14]; 1) Low-risk: clinical stage T1-T2, Gleason score 6 and PSA < 10 ng/ml; 2) Intermediate-risk: T1-T2 and Gleason score 7 or PSA 10–19.9 ng/ml; 3) High-risk or locally advanced: T3, Gleason score 8 and above or PSA 20–49.9 ng/ml; 4) Regional: T4, N1 or PSA 50–499.9 ng/ml; 5) Distant metastases: M1 or PSA ≥ 500 ng/ml.

### Study design

We simulated lifetime trajectories under different scenarios. Each scenario was defined by one of three levels of diagnostic activity; low as in Sweden 1992, intermediate as in Sweden 2016, and high as in Stockholm 2014 (a year when the Stockholm-3 study invited men to PSA testing

[15]), and one of three levels of life expectancy; short for men born in 1912, intermediate for men born in 1952 and long for men born in 1992. In each scenario, 100 repeated simulations of lifetime trajectories of a cohort of 2,000,000 men were performed. Each simulation was summarized over the follow-up period to provide an estimate of the lifetime risk of prostate cancer by calculating the cumulative incidence of prostate cancer by risk category between age 40 and 100. The estimate at age 80, which is the current average life expectancy for men in Sweden, was extracted for tabulation [7]. The cumulative incidence of prostate cancer diagnosis in each risk category was defined as the proportion p = n/N of n diagnosed men with this risk category throughout the simulated follow-up, and the estimate of the variance used to construct confidence intervals was p(1-p)/N. Point and variance estimates extracted from each simulation were pooled across the 100 simulations for each scenario as previously described [8].

The Regional Ethical Review Board of Uppsala University approved the study (approval number 2016–239). Patient consent was collected by opt-out approach.

## Results

### Diagnostic activity

Men exposed to increasingly high diagnostic activity had an increased lifetime risk of prostate cancer diagnosis overall; 18% (95% CI 17%-19%) for low activity, 21% (95% CI 20%-22%) for intermediate, and 29% (95% CI 25%-32%) for high diagnostic activity, applying long life expectancy in all scenarios (**Table 1, Fig 1**). Men exposed to high compared to low diagnostic activity had a five-fold higher lifetime risk of low or intermediate-risk prostate cancer, 22% (95% CI 19%-26%) vs. 5% (95% CI 4%-5%), and half the risk of high-risk or metastatic prostate cancer 13% (95% CI 13%-14%) vs. 6% (95% CI 5%-8%).

### Life expectancy

The lifetime risk of prostate cancer diagnosis overall increased modestly with longer life expectancy increasing from 15% (95% CI 14%-16%) for short life expectancy, to 18% (95% CI 18%-19%) for intermediate, and to 21% (95% CI 20%-22%) for long life expectancy, applying the intermediate diagnostic activity in all scenarios (**Table 2, Fig 2**). Long compared to short life expectancy was associated with a higher lifetime risk of prostate cancer in each risk category; in particular for distant metastases, 3% (95% CI 3%-3%) vs. 1% (95% CI 1%-2%). For a reader-friendly representation of lifetime risk, we also present our results by use of pictograms (**Figs 3 and 4**).

**Table 1. Lifetime-risk of prostate cancer overall and per risk category according to intensity of diagnostic activity.**

| Risk category | Low Sweden 1992 | | Intermediate Sweden 2016 | | High Stockholm 2014* | |
|---|---|---|---|---|---|---|
| | 100 y | 95% CI | 100 y | 95% CI | 100 y | 95% CI |
| All | 17.8 | (16.9–18.8) | 21.4 | (20.4–22.3) | 28.6 | (25.1–32.2) |
| Low-risk | 1.8 | (1.5–2) | 4.8 | (4.4–5.2) | 10.7 | (7.6–13.7) |
| Intermediate-risk | 2.8 | (2.4–3.2) | 7.5 | (6.8–8.2) | 11.6 | (8.2–15.1) |
| High-risk | 5.7 | (5.2–6.2) | 4.4 | (4–4.8) | 2.9 | (2–3.8) |
| Regional metastases | 2.8 | (2.5–3.1) | 1.7 | (1.5–1.9) | 1.4 | (0.9–1.9) |
| Distant metastases | 4.8 | (4.4–5.1) | 3 | (2.7–3.3) | 2.1 | (1.4–2.7) |
| Low or intermediate-risk | 4.5 | (4–5.1) | 12.3 | (11.4–13.2) | 22.3 | (18.8–25.9) |
| High-risk or regional or distant metastases | 13.3 | (12.6–14) | 9.1 | (8.4–9.7) | 6.3 | (4.9–7.8) |

Life expectancy is set to correspond to men born 1992.

*A year when the Stockholm-3 study invited men to measure their PSA [15].

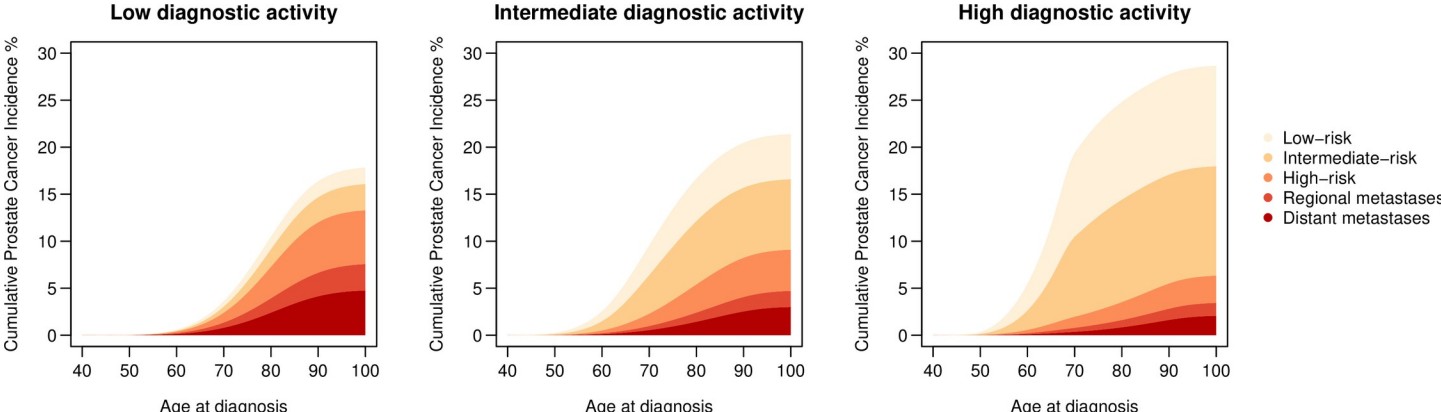

**Fig 1. Cumulative incidence of prostate cancer diagnosis by risk category for a simulated follow-up of 2,000,000 men from age 40 to age 100 under three different levels of diagnostic activity: Low as in Sweden 1992; Intermediate as in Sweden 2016; High as in Stockholm 2014; when the Stockholm-3 study invited men to measure their PSA [15].**

**Table 2. Lifetime-risk of prostate cancer overall and per category according to life expectancy for three birth cohorts.**

| | Short Birth year 1912 | | Intermediate Birth year 1952 | | Long Birth year 1992 | |
|---|---|---|---|---|---|---|
| Risk category | 100 y | 95% CI | 100 y | 95% CI | 100 y | 95% CI |
| All | 14.7 | (13.9–15.5) | 18.4 | (17.6–19.3) | 21.4 | (20.4–22.3) |
| Low-risk | 3.8 | (3.5–4.2) | 4.4 | (4–4.8) | 4.8 | (4.4–5.2) |
| Intermediate-risk | 5.7 | (5.1–6.3) | 6.8 | (6.1–7.4) | 7.5 | (6.8–8.2) |
| High-risk | 2.7 | (2.4–3) | 3.7 | (3.3–4) | 4.4 | (4–4.8) |
| Regional metastases | 1 | (0.8–1.1) | 1.4 | (1.2–1.5) | 1.7 | (1.5–1.9) |
| Distant metastases | 1.5 | (1.4–1.7) | 2.3 | (2.1–2.5) | 3 | (2.7–3.3) |
| Low or intermediate-risk | 9.5 | (8.8–10.3) | 11.2 | (10.3–12) | 12.3 | (11.4–13.2) |
| High-risk or regional or distant metastases | 5.2 | (4.8–5.6) | 7.3 | (6.7–7.8) | 9.1 | (8.4–9.7) |

Diagnostic activity corresponding to 'intermediately high' as in Sweden in 2016.

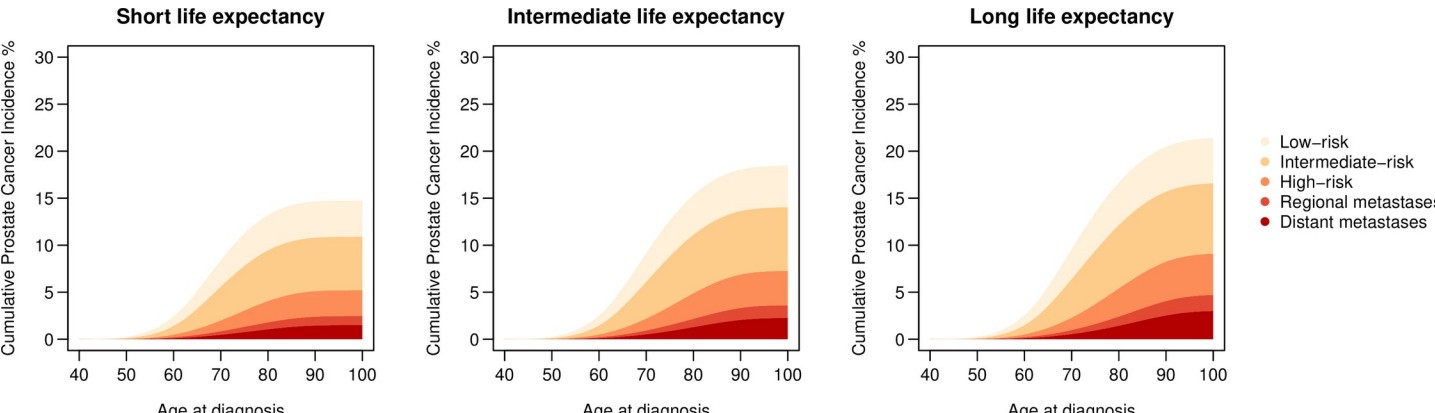

**Fig 2. Cumulative incidence of prostate cancer diagnosis by risk category for a simulated follow-up of 2,000,000 men from age 40 to age 100 under three different levels of life expectancy: Short as for birth cohort 1912; Intermediate as for birth cohort 1952; Long as for birth cohort 1992.**

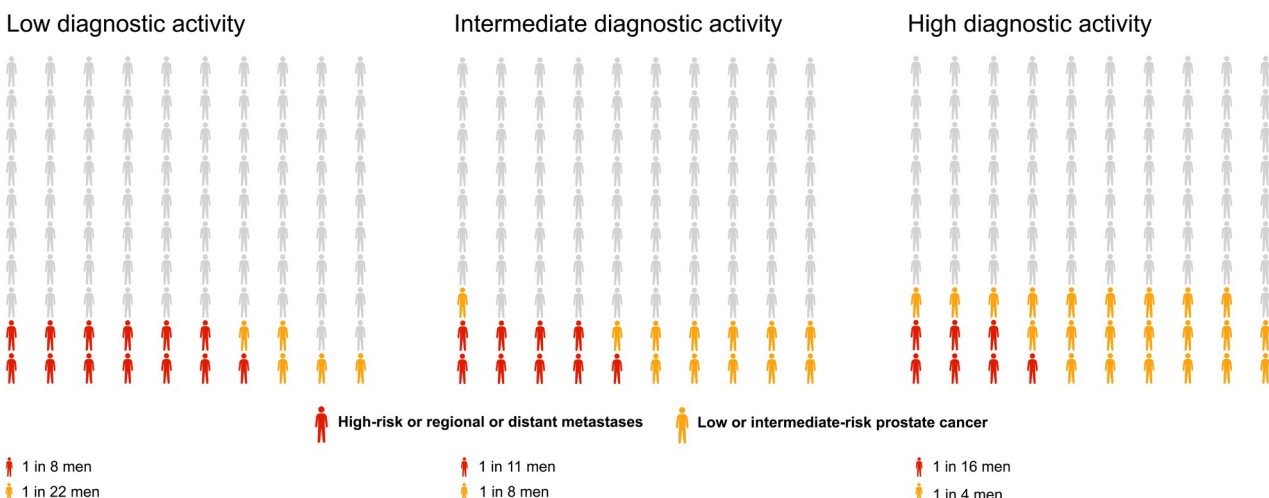

**Fig 3. Lifetime risk of low to intermediate-risk prostate cancer diagnosis (in blue), and high risk or metastatic prostate cancer (in red) under three levels of diagnostic activity: Low as in Sweden 1992; Intermediate as in Sweden 2016; High as in Stockholm 2014; when the Stockholm-3 study invited men to measure their PSA [15].**

Lifetime risk of low-risk prostate cancer increased marginally from age 80 to age 100 for every scenario of diagnostic activity and life expectancy, whereas the lifetime risk of distant metastases doubled for all levels of diagnostic activity for men with a longer life expectancy (**S1** and **S2** Tables).

## Discussion

In this simulation model, lifetime risk of a prostate cancer diagnosis was strongly associated with the level of diagnostic activity and to a lesser degree to potential life expectancy. High compared to low diagnostic activity increased the lifetime risk of low or intermediate-risk prostate cancer five-fold, and halved the risk of high-risk and metastatic prostate cancer,

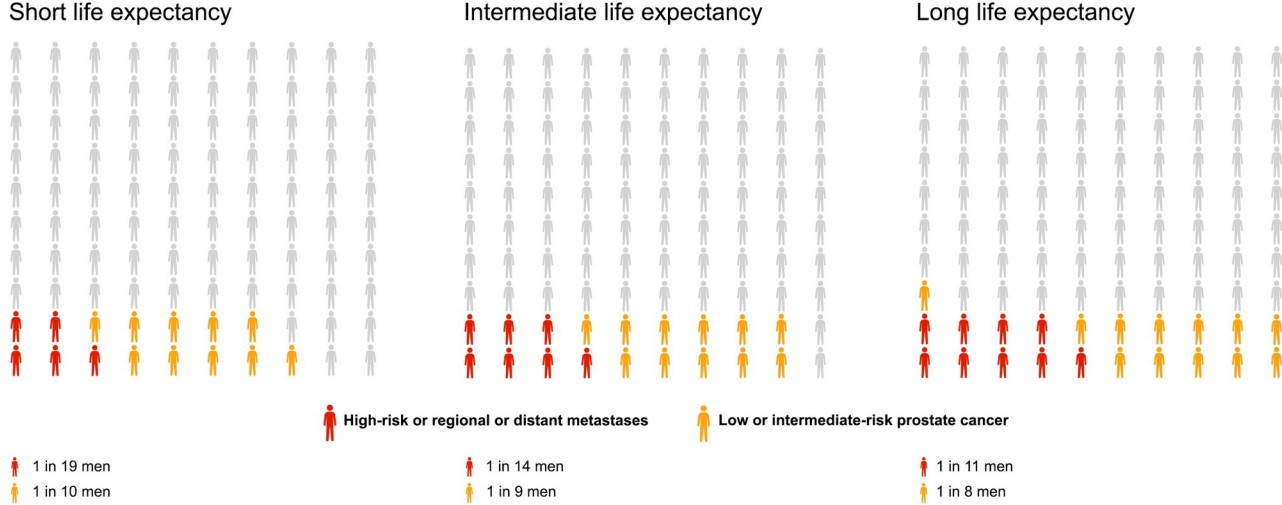

**Fig 4. Lifetime risk of low to intermediate-risk prostate cancer diagnosis (in blue), and high risk or metastatic prostate cancer (in red) under three levels of life expectancy: Short as for birth cohort 1912; Intermediate as for birth cohort 1952; Long as for birth cohort 1992.**

whereas long life expectancy modestly increased the lifetime risk of all prostate cancer risk categories.

The lifetime risk of disease can be estimated in multiple ways. Cumulative risk is commonly utilized for reporting cancer statistics in the Nordic countries (NORDCAN) to calculate the risk of a specific cancer diagnosis from birth until a specified age [16]. The standardized cumulative risk is a useful measure to compare lifetime risk between populations, as it is independent of life expectancy [17]. Since competing risks of death from other causes are not accounted for, the choice of the upper age limit can overestimate lifetime risk. The current probability method described by Goldberg in 1956 is a more accurate estimate of lifetime risk as it accounts for competing risks [18]. This method assumes that all registered cancers are first primaries, which may overestimate the lifetime risk if multiple primary cancers occur. This is not an issue for prostate cancer since death from second primary prostate cancer is rare. PRISM-PC uses the same principles as the cited probability method. Additionally, the model allows for adjustment of diagnostic activity and life expectancy.

In a study from the Surveillance, Epidemiology, and End Results (SEER) Program that applied the current probability method, the lifetime risk of common cancers was calculated using the DevCan software available at the SEER official website [19, 20]. Although African-American men had a higher incidence of prostate cancer than non-Hispanic white men, African-American men had a lower lifetime risk of a prostate cancer diagnosis due to shorter life expectancy. In a study from England the lifetime risk of prostate cancer in black men was 29% compared to 13% in white men, using the same software [21]. In a study from Australia, the lifetime risk of prostate cancer increased by 8% after the introduction of PSA testing [22]. A similar comparison can be made in our study. Men exposed to intermediate compared with low diagnostic activity, corresponding to diagnostic activity after the introduction of PSA testing in Sweden, had a 4% excess lifetime risk of a prostate cancer diagnosis. Ebeling et al. investigated the separate effects of life expectancy and the incidence of disease on the lifetime risk of myocardial infarction and hip fracture [23]. The decrease in the lifetime risk of these events was very small since the decrease in incidence due to preventive measures was counterbalanced by a concomitant increase in life expectancy. In our study long life expectancy and high diagnostic activity both increased the lifetime risk of a prostate cancer diagnosis.

Disease incidence is an important measure to describe population-based disease burden as well as to compare different populations and time periods. In Sweden and the United States, around one third of all men age 60–79 are tested annually for PSA [24, 25]. The uptake of PSA testing has led to a two-fold increase in the age-standardized incidence of prostate cancer, first in the US from 55 to 107 new cases per 100,000 men between 1980–1995 and later in Sweden, from 52 to 111 new cases per 100,000 men between 1990–2005 [26]. This shows that diagnostic activity can drastically change the incidence of prostate cancer in a relatively short period of time.

Disease incidence, however, does not confirm the efficacy of a screening program. As demonstrated in the ProtecT screening and treatment trial, many of the men diagnosed with low-risk prostate cancer are not destined to die from prostate cancer [27]. Conversely a decline in the incidence of metastatic disease associated with higher intensity screening may simply reflect diagnostic lead time. The true value of screening program can only be confirmed by a subsequent decline in prostate cancer mortality which requires data concerning the natural history of screen detected prostate cancer and the efficacy of treatment.

The strengths of our study stems from the validity and completeness of the National Prostate Cancer Register, which includes 98% of all men diagnosed with prostate cancer in Sweden. Our model was based on a comprehensive, highly representative cohort that consisted of virtually all men diagnosed with prostate cancer in Sweden during a 25-year period [28, 29]. Unlike

previous models on prostate cancer that have tried to emulate screening trials, the PRISM-PC model is based on real-world data and relies on not only the uptake of PSA testing but all diagnostic activities leading to a prostate cancer diagnosis. Since no explicit assumptions are made on compliance, test properties, and the natural history of the disease, the model is flexible and allows for changes in incidence, diagnostic activity, and mortality.

A limitation of our model is that it is based on the assumption that risk factors for prostate cancer, other than age, remained unchanged during the simulated follow-up. Changes in these factors may potentially confound the effects of diagnostic activity in the model. However, since there are no known strong risk factors for prostate cancer except age and ethnicity, this should not impact our results [30]. All models are an approximation of reality, however, real-life follow-up data would be impossible to use for the study hypothesis since both diagnostic activity and life expectancy are continuously changing. Modelling techniques are therefore essential to test these factors independently. Another potential limitation of our study is the use of incidence as proxy for diagnostic activity instead of a direct measure of PSA-testing. However, since incidence is the sum of diagnostic activities, including intensity of PSA testing, biopsy frequency, and number of biopsy cores taken, this may in fact not be a disadvantage since an elevated PSA per se does not always lead to subsequent diagnostic work-up.

## Conclusion

The lifetime risk of a prostate cancer diagnosis is strongly influenced by level of diagnostic activity and to a lesser extent by life expectancy. High diagnostic activity leads to an increased lifetime risk of a prostate cancer diagnosis overall, mostly due to a five-fold increased detection of low or intermediate-risk prostate cancer. High diagnostic activity also leads to a two-fold decrease in the incidence of metastatic prostate cancer. Whether this leads to lower prostate cancer mortality or simply reflects diagnostic lead time cannot be determined from our results. Long life expectancy modestly increases the lifetime risk of a prostate cancer diagnosis overall, especially for high-risk and metastatic prostate cancer.

## Supporting information

**S1 Table. Lifetime risk of prostate cancer overall and per risk category according to intensity of diagnostic.**
(DOCX)

**S2 Table. Lifetime risk of prostate cancer overall and per category according to life expectancy for three birth cohorts.**
(DOCX)

## Acknowledgments

This project was made possible by the continuous work of the National Prostate Cancer Register of Sweden (NPCR) steering group: Pär Stattin (chair), Ingela Franck Lissbrant (co-chair), Camilla Thellenberg, Johan Styrke, Hampus Nugin, Stefan Carlsson, David Robinson, Mats Andén, Johan Stranne, Jon Kindblom, Thomas Jiborn, Olof Ståhl, Maria Nyberg, Fredrik Sandin, Karin Hellström, Hans Joelsson and Gert Malmberg.

## Author Contributions

**Conceptualization:** Andri Wilberg Orrason, Marcus Westerberg, Peter Albertsen, David Robinson, Hans Garmo, Pär Stattin.

**Data curation:** Hans Garmo, Pär Stattin.

**Formal analysis:** Andri Wilberg Orrason, Marcus Westerberg, Hans Garmo.

**Funding acquisition:** Pär Stattin.

**Methodology:** Andri Wilberg Orrason, Marcus Westerberg, Peter Albertsen, Hans Garmo, Pär Stattin.

**Project administration:** Pär Stattin.

**Software:** Andri Wilberg Orrason, Marcus Westerberg, Hans Garmo.

**Supervision:** Hans Garmo, Pär Stattin.

**Visualization:** Andri Wilberg Orrason.

**Writing – original draft:** Andri Wilberg Orrason, Pär Stattin.

**Writing – review & editing:** Andri Wilberg Orrason, Marcus Westerberg, Peter Albertsen, Johan Styrke, David Robinson, Hans Garmo, Pär Stattin.

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
