## [Decision Letter · Decision Letter 0]

12 Sep 2022

PONE-D-22-23181Diagnostic activity impacts lifetime risk of prostate cancer diagnosis more strongly than life expectancyPLOS ONE

Dear Dr. Orrason, 

Thank you for submitting your manuscript to PLOS ONE. After careful consideration, we feel that it has merit but does not fully meet PLOS ONE’s publication criteria as it currently stands. Therefore, we invite you to submit a revised version of the manuscript that addresses the points raised during the review process.

ACADEMIC EDITOR:

We look forward to receiving your revised manuscript.

Kind regards,

Donovan Anthony McGrowder, PhD., MA., MSc

Academic Editor

PLOS ONE

3. Our staff editors have determined that your manuscript is likely within the scope of our Early Detection, Screening and Diagnosis of Cancer Call for Papers. This editorial initiative is headed by in-house PLOS editors. This Call for Papers aims to explore recent advances in the early detection of cancer and implications of these advances for patient survival. Additional information can be found on our announcement page: https://collections.plos.org/call-for-papers/early-detection-screening-and-diagnosis-of-cancer/

If you would like your manuscript to be considered for this collection, please let us know in your cover letter and we will ensure that your paper is treated as if you were responding to this call.  Please note that being considered for the Call for Papers does not require additional peer review beyond the journal’s standard process and will not delay the publication of your manuscript if it is accepted by PLOS ONE. If you would prefer to remove your manuscript from collection consideration, please specify this in the cover letter.

Additional Editor Comments:

Dear Dr. Orrason,

Your manuscript “Diagnostic activity impacts lifetime risk of prostate cancer diagnosis more strongly than life expectancy” has been assessed by our reviewers. They have raised a number of points which we believe would improve the manuscript and may allow a revised version to be published in PLOS ONE. Their reports, together with any other comments, are below.

If you are able to fully address these points, we would encourage you to submit a revised manuscript to PLOS ONE.

Regards,

Dr. Donovan McGrowder

Associate Editor

Reviewers' comments:

Reviewer's Responses to Questions

**Comments to the Author**

1. Is the manuscript technically sound, and do the data support the conclusions?

Reviewer #1: Yes

Reviewer #2: No

Reviewer #3: Yes

2. Has the statistical analysis been performed appropriately and rigorously? 

Reviewer #1: Yes

Reviewer #2: No

Reviewer #3: Yes

3. Have the authors made all data underlying the findings in their manuscript fully available?

Reviewer #1: Yes

Reviewer #2: No

Reviewer #3: No

4. Is the manuscript presented in an intelligible fashion and written in standard English?

Reviewer #1: Yes

Reviewer #2: Yes

Reviewer #3: Yes

5. Review Comments to the Author

Reviewer #1: This is a very good paper -- and well-written. The authors have shown convincingly that the lifetime risk of a prostate cancer diagnosis is strongly influenced by diagnostic activity and to a lesser degree by life expectancy. I think this information will be useful to a broad segment of medical practitioners.

Reviewer #2: In this paper, the authors use their newly developed Proxy‐based Risk‐stratified Incidence Simulation Model (PRISM) to validate this obvious truth, and to describe the interplay between age, incidence, mortality, and diagnostic activity in Swedish men between ages 40-100 to quantify the effect of diagnostic activity and life expectancy on the lifetime risk of a prostate cancer (PCa) diagnosis overall and by risk categories. Overall, this is a simulation study, where the factors being simulated have not been described very well.

1. It is quite clear what the risk categories are – and how the patients were classified. However, it is not quite clear how the ‘diagnostic activity’ was quantified. Greater clarity as to how some men were classified as ‘low diagnostic activity’ vs ‘intermediate’ vs ‘high’ is warranted.

2. Similarly, please explain how ‘life expectancy’ was calculated.

3. Instead of simply a simulated follow up, the authors would benefit from comparing the simulation to a real life follow up, and determine whether that real life experience follows that of the simulation.

Reviewer #3: In this manuscript, authors examined the impact of diagnostic activity and life expectancy on the lifetime risk of prostate cancer diagnosis. They employed a state transition simulation model based on Swedish population based data and simulated life trajectories of a large cohort of men from ages 40-100 to estimate the lifetime risk of a prostate cancer diagnosis. Their analyses determined that the lifetime risk of prostate cancer diagnosis is influenced by diagnostic activity. Higher exposure to diagnostic activity resulted in more prostate cancer diagnosis. Men exposed to high diagnostic activity were found to have five fold increased risk of low or intermediate risk prostate cancer and half the risk of high risk prostate cancer as compared to men with low diagnostic activity. Overall, it is a comprehensive study using a large cohort of men with interesting findings.

---

## [Author Response · Author response to Decision Letter 0]

17 Sep 2022

5. Review Comments to the Author

Reviewer #1: This is a very good paper -- and well-written. The authors have shown convincingly that the lifetime risk of a prostate cancer diagnosis is strongly influenced by diagnostic activity and to a lesser degree by life expectancy. I think this information will be useful to a broad segment of medical practitioners.

RE: We thank reviewer #1 for his/her positive feedback.

Reviewer #2: In this paper, the authors use their newly developed Proxy‐based Risk‐stratified Incidence Simulation Model (PRISM) to validate this obvious truth, and to describe the interplay between age, incidence, mortality, and diagnostic activity in Swedish men between ages 40-100 to quantify the effect of diagnostic activity and life expectancy on the lifetime risk of a prostate cancer (PCa) diagnosis overall and by risk categories. Overall, this is a simulation study, where the factors being simulated have not been described very well.

1. It is quite clear what the risk categories are – and how the patients were classified. However, it is not quite clear how the ‘diagnostic activity’ was quantified. Greater clarity as to how some men were classified as ‘low diagnostic activity’ vs ‘intermediate’ vs ‘high’ is warranted.

2. Similarly, please explain how ‘life expectancy’ was calculated.

3. Instead of simply a simulated follow up, the authors would benefit from comparing the simulation to a real life follow up and determine whether that real life experience follows that of the simulation.

RE: We thank the reviewer for valuable remarks. The principles of the PRISM-PC simulation model has previously been thoroughly described (reference 8 and 11). We provide a reference to this methods paper and in addition we also provide a brief description of the model under Methods. We argue that a more detailed review of the model components is beyond the scope of this article and would be of little benefit for the readers of PLOSONE. However, if the editor still wants a more detailed description of the model we are of course willing to provide such a paragraph. 

1. The diagnostic activity is difficult to quantify as the reviewer correctly points out. Diagnostic activity is based on a broad set of factors, such as uptake and intensity of PSA testing, biopsy frequency, number of biopsy cores taken and more recently use of MRT prostate etc. Long time series of such data are not available for the entire Swedish population. However, the result of high diagnostic activity is earlier diagnosis with increased proportion of men diagnosed with low an intermediate-risk prostate cancer. This proportion was therefore used as a proxy for diagnostic activity. Due to the lack of data on the above factors, the proxy is difficult to translate e.g., PSA testing rates. Instead, we use historical and geographical variations in the proxy due to similar variations in the underlying diagnostic activity from real life scenarios. These were based on ours and others qualitative knowledge of the historical diagnostic activity, motivating the names of the scenarios: low as in Sweden 1992, intermediate as in Sweden 2016 and high as in Stockholm 2014. 

The relative differences in diagnostic activity are shown in the figure below for clarification:

Three levels of the proxy for diagnostic activity; low as in Sweden 1992 (green), intermediate as in Sweden 2016 (blue) and high as in Stockholm 2014 (red).

2. The age-conditional life expectancy of men without a prostate cancer diagnosis was calculated using comparator men without a prostate cancer diagnosis in PCBaSe using an age-period-cohort model using age, calendar period and birth cohort data. Age-period-cohort (APC) models are frequently used to estimate life expectancy (Fay et al. Age-conditional probabilities of developing cancer. Statistics in medicine. 2003, Smittenaar et al. Cancer incidence and mortality projections in the UK until 2035. British journal of cancer. 2016).

One of the strengths of our model is that a prostate cancer-free comparator cohort in PCBaSe was used to calculate the life expectancy of men without a prostate cancer diagnosis, whereas most studies are limited to general population life table data which also includes men with prostate cancer. Also, the model allowed us to extrapolate outside of observed calendar periods in order to explore hypothetical improvements in life-expectancy. 

3. Correct. Real-life data would certainly be the ideal choice for our study hypothesis.

However, this is almost impossible to achieve since both diagnostic activity and life expectancy are continuously changing for each birth cohort. The important time horizon is also very long since the risk of prostate cancer becomes non-negligible after age 40 and is elevated throughout the rest of a man’s lifetime. We are not aware of any high-quality data with follow-up that extends that long. Modeling techniques are therefore essential to test these factors independently. To underline this we have added a sentence to the last paragraph in Discussion.

In our previous paper, the output of the simulation model was compared with observed data of incidence and mortality, with good agreement between the two (reference 8). Additional external validation based on data from other countries would be desirable if the appropriate data would be made available e.g., through collaborations.

Reviewer #3: In this manuscript, authors examined the impact of diagnostic activity and life expectancy on the lifetime risk of prostate cancer diagnosis. They employed a state transition simulation model based on Swedish population based data and simulated life trajectories of a large cohort of men from ages 40-100 to estimate the lifetime risk of a prostate cancer diagnosis. Their analyses determined that the lifetime risk of prostate cancer diagnosis is influenced by diagnostic activity. Higher exposure to diagnostic activity resulted in more prostate cancer diagnosis. Men exposed to high diagnostic activity were found to have five fold increased risk of low or intermediate risk prostate cancer and half the risk of high risk prostate cancer as compared to men with low diagnostic activity. Overall, it is a comprehensive study using a large cohort of men with interesting findings.

RE: We thank the reviewer for a positive comment.

---

## [Editor Report · Decision Letter 1]

3 Nov 2022

Diagnostic activity impacts lifetime risk of prostate cancer diagnosis more strongly than life expectancy

PONE-D-22-23181R1

Dear Dr. Orrason,

We’re pleased to inform you that your manuscript has been judged scientifically suitable for publication and will be formally accepted for publication once it meets all outstanding technical requirements.

Kind regards,

Donovan Anthony McGrowder, PhD., MA., MSc

Academic Editor

PLOS ONE

Additional Editor:

Dear Dr. Orrason,

The manuscript entitled “Diagnostic activity impacts lifetime risk of prostate cancer diagnosis more strongly than life expectancy” was revised in accordance with the reviewers’ comments and is provisionally accepted pending final checks for formatting and technical requirements.

Regards,

Dr. Donovan McGrowder (Academic Editor)

---

## [Editor Report · Acceptance letter]

14 Nov 2022

PONE-D-22-23181R1 

Diagnostic activity impacts lifetime risk of prostate cancer diagnosis more strongly than life expectancy 

Dear Dr. Orrason:

I'm pleased to inform you that your manuscript has been deemed suitable for publication in PLOS ONE. Congratulations! Your manuscript is now with our production department. 

Kind regards, 

on behalf of

Dr. Donovan Anthony McGrowder 

Academic Editor

PLOS ONE